# Integration of Mutational Signature Analysis with 3D Chromatin Data Unveils Differential AID-Related Mutagenesis in Indolent Lymphomas

**DOI:** 10.3390/ijms222313015

**Published:** 2021-12-01

**Authors:** Julieta H. Sepulveda-Yanez, Diego Alvarez-Saravia, Jose Fernandez-Goycoolea, Jacqueline Aldridge, Cornelis A. M. van Bergen, Ward Posthuma, Roberto Uribe-Paredes, Hendrik Veelken, Marcelo A. Navarrete

**Affiliations:** 1Department of Hematology, Leiden University Medical Center, 2300 RC Leiden, The Netherlands; j.h.sepulveda_yanez@lumc.nl (J.H.S.-Y.); C.A.M.van_Bergen@lumc.nl (C.A.M.v.B.); J.H.Veelken@lumc.nl (H.V.); 2School of Medicine, University of Magallanes, Punta Arenas 6210427, Chile; diego.alvarez@umag.cl; 3Centro Asistencial Docente y de Investigación, University of Magallanes, Punta Arenas 6210005, Chile; 4Department of Physics and Mathematics, University of Magallanes, Punta Arenas 6210427, Chile; jose.fernandezg@umag.cl; 5Department of Computer Engineering, University of Magallanes, Punta Arenas 6210427, Chile; jacqueline.aldridge@umag.cl (J.A.); roberto.uribe@umag.cl (R.U.-P.); 6Department of Oncology, Reinier de Graaf Hospital, 2625 AD Delft, The Netherlands; e.posthuma@rdgg.nl

**Keywords:** follicular lymphoma, chronic lymphocytic leukemia, mutational signatures, activation-induced cytidine deaminase, DNA repair pathways

## Abstract

Activation-induced deaminase (AID) is required for somatic hypermutation in immunoglobulin genes, but also induces off-target mutations. Follicular lymphoma (FL) and chronic lymphocytic leukemia (CLL), the most frequent types of indolent B-cell tumors, are exposed to AID activity during lymphomagenesis. We designed a workflow integrating de novo mutational signatures extraction and fitting of COSMIC (Catalogue Of Somatic Mutations In Cancer) signatures, with tridimensional chromatin conformation data (Hi-C). We applied the workflow to exome sequencing data from lymphoma samples. In 33 FL and 30 CLL samples, 42% and 34% of the contextual mutations could be traced to a known AID motif. We demonstrate that both CLL and FL share mutational processes dominated by spontaneous deamination, failures in DNA repair, and AID activity. The processes had equiproportional distribution across active and nonactive chromatin compartments in CLL. In contrast, canonical AID activity and failures in DNA repair pathways in FL were significantly higher within the active chromatin compartment. Analysis of DNA repair genes revealed a higher prevalence of base excision repair gene mutations (*p* = 0.02) in FL than CLL. These data indicate that AID activity drives the genetic landscapes of FL and CLL. However, the final result of AID-induced mutagenesis differs between these lymphomas depending on chromatin compartmentalization and mutations in DNA repair pathways.

## 1. Introduction

Cancer, that is, the uncontrolled proliferation of transformed cells, is generally attributable to acquired or inherited genetic variants that affect crucial cellular pathways [1]. Acquired mutations can be caused by environmental influences or stochastic errors in DNA replication [2,3]. Particular mutagenic mechanisms generate distinguishable mutational signatures across a cancer cell’s genome [4,5].

Somatic hypermutation (SHM) represents an endogenous mutator mechanism in B lymphocytes. Physiologically, SHM targets immunoglobulin genes (IG) and is dependent on the expression of activation-induced deaminase (AID) in germinal center reactions. AID induces the deamination of deoxycytidine into deoxyuridine [6]. Subsequent activation of DNA repair mechanisms may result in faithful base repair but may alternatively lead to genetic variants, particularly through the engagement of the error-prone alternative base excision repair (BER) and mismatch repair (MMR) pathways [7,8].

AID-induced SHM targets preferentially distinct sequence motifs: Canonical C>T/G transitions occur in WRCY motifs [9], and non-canonical A>C transversions in WA motifs [10]. A third pattern, it is characterized by C>T transitions in RCG motifs [11]. Aberrant AID activity on non-IG target genes has been implicated in the pathogenesis of various types of lymphoma [12,13]. However, the genome-wide consequences of AID-associated mutational signatures have not yet been analyzed specifically per type of B-cell lymphoma.

In the genome-wide context, the three-dimensional (3D) chromatin structure could play an important role in the activity of the different mutational mechanisms. The introduction of High-throughput Chromosome Conformation Capture (Hi-C) [14] allows the identification of different states of the genome structure at the sub-chromosomal scale. Genomic regions can be assigned according to Hi-C to two compartments: the active compartment (compartment A) includes genomic regions characterized by transcription or epigenetics marks associated with open chromatin (H3K36me3), high density of genes, and DNase I hypersensitivity [14]. In contrast, the inactive compartment (compartment B) represents the condensed DNA regions [14,15]. Hi-C maps from pro-B cells revealed that up to 96% of canonical AID target regions can be assigned to compartment A in the mouse genome [12].

Follicular lymphoma (FL) and chronic lymphocytic leukemia/small lymphocytic lymphoma (CLL) are the most frequent types of indolent B-cell neoplasia [16]. Consistent with a malignancy of germinal center B-cells, FL cells constitutively express AID, and FL cells acquire very high levels of SHM in their IG genes. AID expression in FL can be correlated to both physiological and aberrant SHM [17,18]. In contrast, CLL cells do not reside in germinal centers. Nevertheless, a subset of CLL has been exposed to AID activity as indicated by modest SHM of their immunoglobulin genes [19,20]. Our aim was to identify AID-induced mutagenesis depending on chromatin compartmentalization in FL and CLL. We analyzed the contribution of AID to mutational signatures in whole-exome sequencing data of FL and CLL cases and the relationship between these signatures and the B-cell chromatin structure.

## 2. Materials and Methods

### 2.1. Patient Characteristics and Sample Acquisition

Cryopreserved viable cells from 15 blood, bone marrow, and lymph node samples with infiltration by FL (*n* = 9), CLL (*n* = 3), or CLL-phenotype monoclonal B lymphocytosis (*n* = 3) were obtained from the biobank of the LUMC Department of Hematology. This study was conducted in accordance with the Helsinki Declaration, all samples had written informed consent, and the study was performed with IRB approval of the local ethics committee (no. B16.039). Single cell suspensions were obtained by gentle mechanical disruption and mesh filtration and were cryopreserved using 10% DMSO as cryoprotectant. The remaining tissue was cultured in low-glucose DMEM to obtain stromal cell cultures for isolation of germinal DNA from nonmalignant cells. Thawed single cells were purified by flow cytometry using fluorescently labeled antibodies specific for CD19 and CD10 for FL, and CD19, CD5 and CD20dim for CLL and rested overnight followed by removal of dead cells using MACS dead cell removal kit. Additional whole-exome sequencing (WES) data were obtained for 24 FL and 24 CLL samples and their germ-line reference from European Genome-phenome Archive (https://ega-archive.org, (accessed on 30 November 2021)) [21] as provided by Barts Cancer Institute, London (EGAD00001001301) [22], and from Institute Gustave Roussy, Villejuif (ERP003635) [23]. Patients’ characteristics are shown in Appendix A.

#### 2.1.1. Library Preparation and Sequencing

Genomic DNA was isolated using QIAamp DNA Mini kit (Qiagen, Hilden, Germany). Samples were sequenced (HiSeq 4000 instrument, Illumina, San Diego, CA, USA) in paired-end mode on Illumina (2 × 101 bp) using TrueSeq DNA exome kit (v.6) (Illumina, San Diego, CA, USA). Mean coverage for every tumor and germline sample is depicted in Appendix A.

#### 2.1.2. Sequence Alignment and Variant Calling

Paired-end reads were aligned to the human reference genome sequence GRCh38 using BWA–MEM (V0.715-r1140) [24]. Alignment metrics and insert size distribution were gathered specifically through the CollectAlignmentSummaryMetrics and CollectInsertSizeMetrics tools from Picard (v2.12.1) [25]. Duplicate fragments were marked and removed using Picard (v2.12.1) tool MarkDuplicates. Local realignment was performed around indels to improve SNP calling in these conflictive areas with IndelRealigner tool. To avoid recalibration biases that might affect samples independently of each other, base quality scores were recalibrated using the BaseRecalibrator tool, with standard hard filtering parameters or Variant Quality Score Recalibration (VQSR) according to Genome Analysis Toolkit (GATK) [25] Best Practices.

Variant calling was performed on mpileup output files using Varscan (V2.3.9) [26] to WES data from tumor and patient-matched normal samples with a minimal variant frequency of 0.2, a somatic *p*-value of 0.05, and minimum coverage of 10×. Filtered variants were then annotated applying Annovar (v.2016Feb01) [27] based on versions of the 1000 Genomes Project (2015 Aug) [28], the Exome Aggregation Consortium (ExAC) [29], and predictions of functional importance from SIFT (Sorting Intolerant From Tolerant) [30] and by applying LRT (Likelihood Ratio Test) [31]. Variants were filtered and associated with DNA repair pathways by the genes related to BER, MMR, Fanconi anemia (FA), and DNA damage response (DDR) including their known variants. Quality control metrics were assessed using FastQC (v0.11.2) [32] before and after the alignment workflow and were reviewed to identify potential low-quality data files (Figure 1).

#### 2.1.3. Mutational Signatures Analysis

1.Variant classification in 96 contexts: Every single nucleotide variant (SNV) was represented for example, as C>A, C>T (by convention beginning with the pyrimidine) within a trinucleotide context (e.g., “GCT>GAT”). This approach yields 96 different possible contexts (six substitution type C>A, C>G, C>T, T>A, T>G, T>C with four possible bases immediately 5’ and 3’ to each substitution). This classification annotation was performed for mutations within localized regions (IG loci) as well as for genome-wide mutations (WES) (Figure 1).2.Signature extraction, similarity and fitting: Global and localized mutational signatures were defined by a workflow encompassing a three-step procedure, starting with a de novo signature extraction, followed by an similarity analysis, to allow a final fitting approach. de novo mutational signature extraction was generated by a non-negative matrix factorization (NMF) using R-package SigProfiler [33]. One of the practical drawbacks of the multiplicative NMF algorithm is that the task of selecting the appropriate number of sources is left to the user. Using this tool automatically allows the identification of the optimal number of operative signatures in our dataset (Figure 1, Appendix A).

The similarity between de novo extracted signatures and the consensus Single Base Substitution (SBS) signatures deposited in the COSMIC catalog (v3.2–March 2021) [34,35] was measured by non-negative least squares (NNLS) from SciPy python library (Appendix A) [36]. Finally, signature fitting was performed to define the contribution of the matching signatures to every sample by deconstructSigs [37] (Figure 1).

#### 2.1.4. Unsupervised Clustering Analysis

The relative contribution of the identified de novo mutational signature and signatures from COSMIC identified in the fitting approach (SBS1, SBS3, SBS5, SBS6, SBS9, SBS84) were assembled for all FL and CLL/MBL samples in a matrix and tested for clustering tendency by two methods. The first method was a principal component analysis and the second method was a Hopkins’ test to test the spatial randomness of the data (H = 0.28) [38]. Subsequently, all samples of the matrix were analyzed by divisive hierarchical clustering (DIANA) [39]. Strengths of clustering and estimation of average distances between clusters were analyzed by silhouette analysis (average silhouette width: 0.35, Appendix A) [40] (Figure 1).

#### 2.1.5. Hi-C Data Analysis and Compartment Identification

Compartment identification was performed using previously published Hi-C matrices [15] from lymphoblastoid B cell lines (GM12878) by the neural network based tool SNIPER [41], using the 10% of the data for training (Figure 1). The obtained compartments were used to annotate each variant using Vcfanno [42].

## 3. Results

### 3.1. Variant Allele Densities and Frequencies in FL and CLL

WES data of 33 FL and 27 CLL and 3 MBL tumor samples and matched germline DNA from our own and previously reported [22,23] samples were analyzed by Varscan to detect somatic single nucleotide variations (SNVs) present in at least two reads in a tumor sample. For tumor samples, the average on-target rate was 84.1% (range: 82.3–86.1%), the average depth of coverage was ×133.5 (range: 119.7–147.4) with an average coverage rate over ×10 of 95.3% (range: 94.2–96.5%). For germline samples the average on-target rate was 82.7% (range: 80.6–84.8%), the average depth of coverage was ×93.8 (range: 78.0%–109.6%) with an average coverage rate over 10× of 92.9% (range: 90.7–95.1%) (Appendix A).

The application of a homogeneous streamlined pipeline for all samples provided us with the appropriate mutation metrics to perform a direct comparison. In FL cases, the median number of bona fide somatic mutations was 285 per tumor (range: 248.8–309.6), corresponding to a median mutation rate of 5.8 per Mb (range: 5.1–6.3). Since MBL was indistinguishable from CLL in all analyzed metrics, MBL and CLL cases are grouped under the same category. With a median of 179 mutations per tumor (range: 150.6–207.8) and a median mutation rate of 3.5 per Mb (range 3.1–4.2), CLL/MBL cases carried significantly fewer mutations than FL (Figure 2A), this is also observed in the CLL mutated and unmutated groups (Appendix A). In addition, the median variant allelic frequency of tumor samples was significantly higher in CLL/MBL as compared to FL (Figure 2B); this is preserved in the CLL mutated and unmutated groups (Appendix A).

### 3.2. Somatic Mutations in CLL/MBL and FL Are Frequently Associated with a Deamination Pattern in AID Motifs

The overall pattern of nucleotide substitutions was virtually identical in FL and CLL/MBL (Figure 3A) and was dominated by transitions over transversions with a transition/transversion ratio of 1.6. C>T (G>A) transitions were overrepresented and comprised 37.6% of all mutations. Moreover, 35.6% of all C>T substitutions (13.4% of all somatic mutations) were observed at CpG sites. Since methylated cytosine can easily undergo the transition to thymine [43], C>T transitions at CpG sites are generally accepted to occur through a deamination event. The second most frequent substitutions (23.7%) were T>C transitions, possibly originating from direct oxidation of thymine, or after replication of substrates containing 5-hydroxymethyl or 5-formyl-uracil [44]. Considering the observed dominance of deamination-related mutations, we then analyzed whether the mutations were associated with any of the reported AID motifs WRCY [9], WA [10] and RCG [11]. Indeed, large fractions of the mutations in both FL (42.7%) and CLL/MBL (33.6%) occurred at these AID motifs. The non-canonical WA motif alone contributed 24.7% of the mutations in FL and 18.1% in CLL/MBL. Mutations of the canonical WRCY contributed 8.8% of FL mutations and 6.7% of CLL/MBL mutations. Finally, 9.3% of mutations in FL and 8.8% in CLL/MBL involved the RCG motif. To define how frequently AID motifs would be affected by mutagenesis in human cancer without direct evidence for AID-induced mutagenesis, we analyzed a dataset of 264 samples of cutaneous melanoma and 791 samples of breast cancer The Cancer Genome atlas (TCGA) [45]. The three AID motifs combined contributed only 9.2% of the total number of mutations in cutaneous melanoma (SkCM) and 21% in breast cancer (BRCA) (FL vs. SkCM *p* < 0.0001, FL vs BRCA *p* < 0.0001, CLL/MBL vs. SkCM *p* < 0.0001, CLL/MBL vs SkCM *p* < 0.0001, Pearson’s Chi-squared test) (Figure 3B). The contribution of mutations to these motifs in other cancer types listed in TCGA [45] is shown in Appendix A.

### 3.3. Trinucleotide Context of Somatic Mutations in FL and CLL/MBL

All nucleotide substitutions were assigned to 96 substitution patterns according to their trinucleotide context. While FL and CLL/MBL had a similar pattern overall, FL showed a more frequent contribution of mutation in contexts associated with the canonical (c-AID) and non-canonical (nc-AID) AID motifs than CLL/MBL (Figure 3B, FL: 33.43% and CLL: 24.77%). Since AID activity is expected to preferentially target immunoglobulin genes, we analyzed the mutational pattern restricted to these loci. Within these regions, the 96 substitutions pattern was closely related to the canonical AID signature (Appendix A). This is supported by cosine similarity between the mutational pattern observed at the Ig loci and the signature SBS84 (FL: 0.79 and CLL/MBL: 0.78, Appendix A).

### 3.4. Inferring the Role of AID as an Underlying Mutagenic Mechanism in FL and CLL/MBL

We performed a de novo signature extraction using the 96-trinucleotide context catalog [5]. Since the number of signatures that can be accurately extracted depends on the number of samples and the variance (number of mutations) of the dataset, we calculated the optimal number of signatures using a multiplicative NMF approach recently published [34].

In this unsupervised analysis, the mutational spectrum variance was explained by three signatures (Figure 4A,B, Appendix A). The consistency and stability of the signature extraction was confirmed by repetitive extractions with bootstrapping strategies of sample subsets including categorization by neoplasm type and treatment status. Under the different tested condition, we obtained a consistent and reproducible signature extraction (Appendix A). The global signature analysis was followed by a localized de novo signature extraction on the IG loci, where two signatures were identified (Appendix A).

Next, we analyzed whether the extracted signatures corresponded to known mutational processes described in the COSMIC catalog (v3.2—March 2021) [34,35]. The first signature (germinal center: GC) had a unique composition, the second signature could be attributed to the combination of mutational processes SBS1 and SBS5 (cosine similarity SBS1+SBS5 = 0.91), and the last signature was related to the combination of processes SBS3 and SBS6 (cosine similarity SBS3 + SBS6 = 0.85, Appendix A). As expected for the IG localized de novo signatures, the signatures represented the combination of the recently described signatures SBS84/SBS85 and also SBS37 [34] (Appendix A).

To identify a per sample signature contribution, we used the signature fitting approach [46], incorporating biologically relevant signatures from the COSMIC catalogue (SBS1, SBS3, SBS5, SBS6, SBS9, SBS84) as well as the new de novo signature (GC). We also performed divisive hierarchical clustering (DIANA) for unsupervised sample classification, using seven mutational signature contributions, this analysis yielded three clusters (Figure 4C). FL cases were present mainly in two clusters (II and III in Figure 4C). Cluster two was dominated by SBS3 and SBS5 (mean contribution of SBS3 + SBS5: 61.0%). In contrast, the new GC signature contributed prominently to cluster three (mean contribution of GC signature: 63.0%). CLL cases were predominately allocated to cluster one with a dominance of SBS5 (mean contribution SBS5 in cluster 1 is 65.6%). Despite the dominance of AID-related mutations in the genomic landscapes of both FL and CLL/MBL, these data suggest that intrinsic differences in mutational signatures exist between these types of B-cell lymphoma.

### 3.5. FL and CLL Show Differential Distribution of Mutational Signatures across Tridimensional (3D) DNA Compartments

To evaluate whether the previously identified signatures have a differential contribution at the chromatin level, we used Hi-C data from B-cells [15] to allocate variants to the active nuclear compartment A and the inactive compartment B. We observed a stable distribution of the signatures across the nuclear compartments in CLL, and that distribution was independent of their mutational status (Figure 5A, ns). However, in FL cases, signatures associated with the DNA mismatch repair process (SBS6) were dominant in the active compartment A, and the signature related to spontaneous deamination (SBS1) in the inactive compartment (Figure 5A, Wilcoxon test with Bonferroni correction: 0.0003 and 0.0002 respectively). The relative contribution of SBS84 associated with mutations in the canonical AID motif (RCY, Figure 4B) was significantly higher in the active compartment (Figure 5A, Wilcoxon test with Bonferroni correction: 0.0009), whereas mutations associated with the non-canonical AID motif (WA) defined by SBS9 (Figure 5B) showed a significantly higher contribution within the inactive compartment (Figure 5A, Wilcoxon test with Bonferroni correction: 0.0024).

### 3.6. Analysis of Mutations in Genes Involved in DNA Repair

Whereas AID initiates mutagenic events by creating U:G mismatches, DNA repair mechanisms subsequently execute definite DNA alterations. Therefore, we searched for evidence of alterations in DNA repair pathways that could explain the observed difference in the substitution patterns and signatures compartment between FL and CLL/MBL. The list of genes involved in the DNA repair pathways was defined by the KEGG database and literature [47]. We found novel mutations (i.e., variants not present in germ-line or a preceding biopsy) in one or more DNA repair pathways in 23 of 33 FL samples (69.7%) and in 14 of 30 CLL/MBL samples (46.7%) (Figure 6A). In FL, the most frequently mutated DNA repair pathway (21 variants in 12 cases) was Fanconi anemia (FA), in particular in the gene FANCD2 with even several mutations in the same patient (Appendix A). The second frequently affected pathway in FL was BER (11 variants in 10 samples) with mutations in its main components, POLE and UNG. With a single variant found in the CLL/MBL cases, the prevalence of BER mutations in FL was significantly higher than in CLL (Fisher´s exact test with Benjamini-Hochberg correction: *p* = 0.026; Figure 6B, Appendix A). In CLL, the most frequently mutated pathway was the DNA damage response (DDR), with a total of 11 variants in 10 samples but without significant difference as compared with FL (Figure 6C). The main DDR genes mutated in CLL were TP53, ATM, and HUS1 (Appendix A).

## 4. Discussion

Sequential acquisition of genomic alterations is considered to play a central role in oncogenesis and tumor progression. The underlying mechanisms of the mutagenic mechanisms continue to be intensively investigated and can be inferred by the analysis of mutational signatures. Here, we report the relative contribution of contextual somatic variants and mutational signatures in relationship with the genome chromatin structure for the two most prevalent indolent mature B-cell neoplasms—FL and CLL.

While the genomic landscape of FL is characterized by a higher mutation density and mutation heterogeneity than CLL, the dominant genomic changes in both types of B-cell lymphoma are C>T transitions, indicating that deamination plays a central role. In both FL and CLL, an important number of somatic variants (42% and 33%, respectively) are located in DNA contexts related to AID motifs. This key observation is in remarkable contrast to the 9% and 21% observed in cutaneous melanoma and breast cancer and provides independent support for the postulated role of AID in FL and CLL development and progression [17,19]. In FL, more mutations have been described in motifs recognized by AID and by APOBEC at relapse than that at diagnosis, and FL patients with accumulated mutations in AID-targeted genes are at high risk for transformation [48]. Besides, a recent study demonstrated that FL tumors harbor excess mutations in AID-motif overlapping the CpG methylation site [11]. Our data indicate that this finding may also extend to CLL.

Underlying mutational processes that shape cancer genomes have been identified by deconvolution of mutations classified by their context of neighboring 5’ and 3’ nucleotides. A set of consensus signatures has been linked to distinct mutagenic mechanisms such as aging, tobacco smoking, deficient mismatch repair, and UV light [4,34,49]. One of the main advantages of deconvolution by NMF algorithms [50] is the lack of bias and independence on external information to extract mutational signatures. Our signatures are reliably related to FL and CLL as indicated by its consistent emergence with bootstrapping strategies. Signatures are usually complex, and they may result from a mix of different mechanisms [46]. In fact, one of the extracted signatures reflects a combination of defects in homologous recombination (SBS3 from COSMIC catalog) and DNA mismatch repair (SBS6 from COSMIC catalog) [4,34]. The next extracted a signature correlated with the ubiquitous processes SBS1, which is associated with spontaneous deamination, and SBS5, which is present in almost any cancer type [4,34,51]. The third novel GC signature was characterized by a distinctive pattern with remarkable dominance of contextual somatic variants in AID motifs and was dissimilar to known COSMIC signatures (cosine similarity < 0.85).

The recently reported signatures, SBS84 and SBS85, which are related to canonical AID activity [34], were difficult to extract from global genomic data. However, these signatures can be readily detected by localized extraction in the IG loci (Appendix A). This apparent discrepancy indicates that the identification of certain signatures, despite their undisputed presence in the dataset, remains challenging by applying only the de novo signature extraction method and requires the addition of a fitting approach. Of note, the signature SBS9, described as related to non-canonical AID activity [52], did not emerge within global or localized extraction.

A signature analysis performed on the basis of the relative contribution of all signatures discussed above, that is, GC, SBS1, SBS3, SBS5, SBS6, SBS9, SBS84 across all analyzed FL and CLL cases, yielded three clusters that were clearly related to both lymphoma types. The cluster goodness is presented in Appendix A. In the first cluster, whose genomic landscape was dominated by a so-called flat signature (SBS5), 81.5% of the cases (22/27) were CLL. Cases in the second cluster showed a higher proportion of AID-related signatures (SBS9 and SBS84), and 63.6% (13/22) were FL cases. The third cluster, strongly dominated by the GC signature, had only FL cases (14 cases, 100%). These data indicate that FL and CLL share common endogenous mutagenic processes during lymphomagenesis, but additional mechanisms influence the activity and downstream consequences of AID in a lymphoma subtype-specific manner.

Since enzymatic deamination is more likely to occur in super-enhancer domains [12,53] and since up to 96% of AID targets may be restricted to the active chromatin compartment [12], we also analyzed the contribution of particular signatures across genome compartments by integrating conformational information obtained from Hi-C data [15]. While no significantly differential contribution of signatures between compartments was observed in CLL, the canonical AID signature (SBS84) and the mismatch repair-related signature (SBS6) provided significantly higher contributions to the genomic landscape of the active compartment of FL cases. These data globally quantify the direct mutagenic effect of ongoing AID activity in FL as corroborated by the ongoing, constitutive AID overexpression in FL as opposed to CLL [17,18,54].

The Pol-*η* related signature (SBS9), that has been linked to non-canonical AID activity and CLL pathogenesis [52], as well as the spontaneous deamination-related signature SBS1 [34], were higher within the inactive compartment of FL. This finding may be indicative of the broader effects of these mutagenic mechanisms or may be reflecting mutations acquired through earlier events occurring in a mutation-prone environment such as the germinal center, and needs further investigation. Because mutagenesis and chromatin conformation are dynamic processes, future systematic analyses over time may be desirable to refine the results of this study. Nevertheless, the folding patterns of chromatin compartment domains are highly conserved within B-cells, and even during malignant B-cell transformation, gene switching from the active to the inactive compartment was only 3.1% [55]. Whether specific chromatin organizations affecting CLL and FL may alter our findings, remains unanswered and will require technically challenging Hi-C experiments on human primary lymphoma cells. On the other hand, the limited number of available samples may restrict broader extrapolation of our results; however, the sample size remains similar to that of other studies addressing mutational signatures analysis [20,46].

Although AID activity can initiate the mutagenic process, different DNA repair pathways define the outcome of the mutational cascade initiated by deamination. Mutations in these pathways can lead to an increased susceptibility to different cancers, such as diffuse large B-cell lymphoma, myeloid leukemia, breast cancer, and to cancer treatment-related toxicity [56,57,58]. We explored the novel hypothesis that acquired mutations in DNA repair pathways might contribute to differential mutational signatures according to lymphoma type. Indeed, a high prevalence of novel mutations was found in at least one pathway in both CLL and FL. Mutations predominantly affected FA and BER pathways in FL and DDR in CLL/MBL. A significantly higher incidence of BER pathway mutations occurred in FL and suggest the investigation of the association of DNA repair pathway mutations with AID-dependent mutational signatures in future studies.

## 5. Conclusions

In summary, we defined the mutational processes that shape the mutational landscape of FL and CLL (global and localized) and integrated these signatures with sub-chromosomal conformation data. As indolent B-cell malignancies, CLL and FL share a common background of mutational processes. In CLL, mutational signatures are evenly distributed across chromatin compartments. In contrast, mutagenesis related to canonical AID activity and failures in DNA repair pathways in FL were more frequently found in the active chromatin compartment. The constitutive AID expression observed in FL and mutations in DNA repair pathways are candidate factors to explain these lymphoma-specific differences. Collectively, these new findings support a direct association between aberrant AID action and lymphomagenic mutations. Since certain mutators, such as ongoing endogenous deamination, are more prone to occurring in restricted areas of the tridimensional structure, the integration of genomic conformational data into signature analysis could help us to better understand the biological relevance of deconvoluted mutational processes.

## Figures and Tables

**Figure 1 ijms-22-13015-f001:**
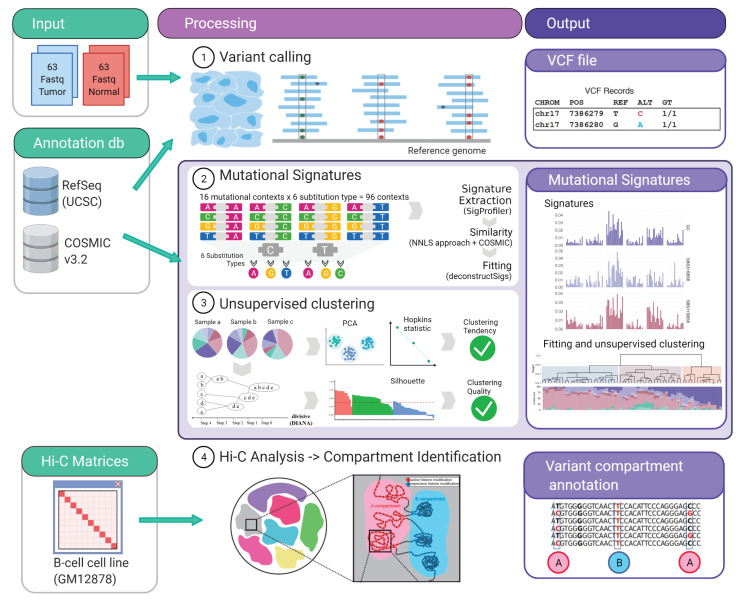
Schematic pipeline workflow. The input data are indicated in green boxes. The middle panel show the different analysis performed and the outputs are indicated in purple boxes. Figure created with BioRender.com.

**Figure 2 ijms-22-13015-f002:**
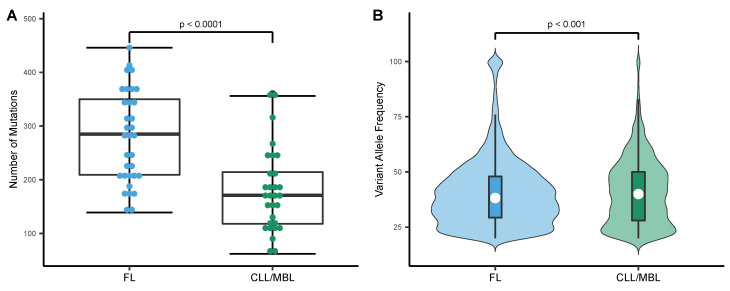
Mutational load and distribution of variant allele frequency in FL and CLL/MBL. (**A**) The mutation rate expressed as number of mutations per exome was higher in FL (*n* = 33) than in CLL/MBL (*n* = 30) (*t*-test, two-sided). (**B**) Violin plots depict the distinct distribution of variant allele frequency in FL and CLL/MBL. White circle: Median variant allele frequency; Colored bars: 25th and 75th percentiles; Whiskers: 5th and 95th percentiles (Wilcoxon test).

**Figure 3 ijms-22-13015-f003:**
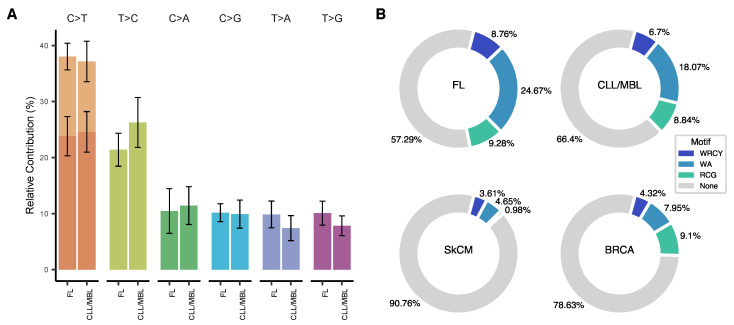
Type of substitutions and mutational patterns in FL and CLL/MBL. (**A**) The base substitution types across FL and CLL/MBL cases are dominated by transitions. (**B**) When tracing AID motifs in FL and CLL/MBL a high proportion of somatic mutations are allocated in such motifs, in strong contrast with skin cutaneous melanoma (SkCM) and Breast cancer (BRCA) that served as non-lymphoid malignancy references.

**Figure 4 ijms-22-13015-f004:**
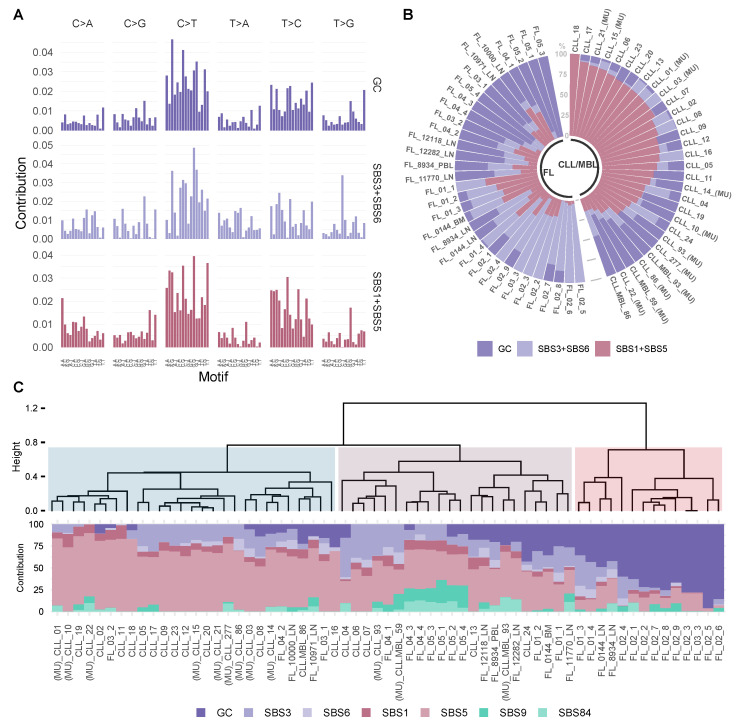
Mutational signatures analysis reveals a stable contribution of signature GC in FL. (**A**) The dataset was explained by 3 mutational signatures, a newly emerged de novo signature GC (GC) and two combinations of signatures derived from COSMIC, Single base substitution signature 3 and Single base substitution signature 6 (SBS3 + SBS6) and Single base substitution signature 1 and Single base substitution signature 5 (SBS1 + SBS5). (**B**) The prevalence of these signatures in individual tumor samples is depicted in each bar and represents an individual exome. (**C**) Fitting using de novo and COSMIC signatures. At the top an unsupervised divisive hierarchical clustering (DIANA) based on the matrix of signature contribution per sample, shows the classification of most CLL/MBL cases distant from FL. The samples with (MU) indicate CLL/MBL cases with mutated IGHV.

**Figure 5 ijms-22-13015-f005:**
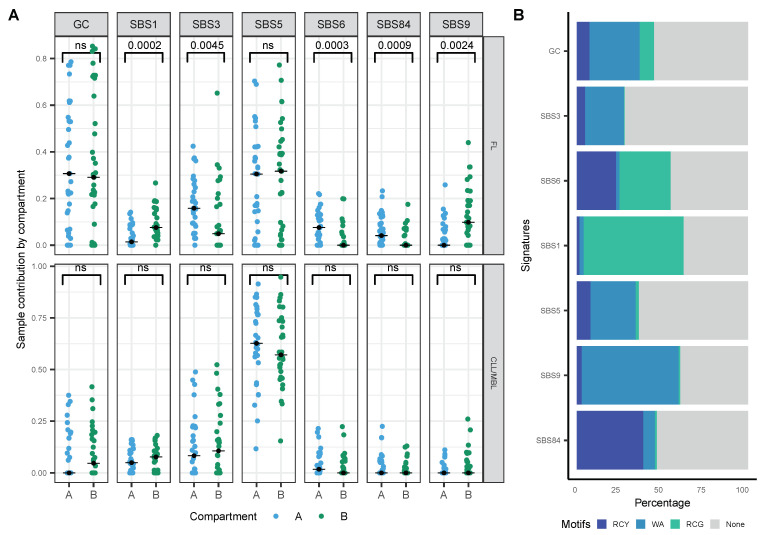
Contribution of mutational signatures on the three-dimensional (3D) chromatin structure. (**A**) Sample contribution by compartment A and B in FL and CLL/MBL cases (Wilcoxon test with Bonferroni correction), black dot: median. (**B**) Relative contribution of AID mtifs (trinucleotide) in signatures analyzed.

**Figure 6 ijms-22-13015-f006:**
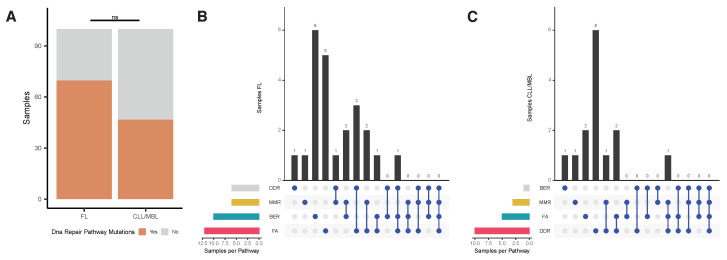
Analysis of mutations in DNA repair pathways. (**A**) Percentage of samples with at least one novel mutation (variants not present in germ-line or a preceding biopsy) in the four DNA repair pathways (Base Excision repair [BER], Mismatch Repair [MMR], Fanconi Anemia pathway [FA], and DNA Damage Response [DDR]) analyzed in FL and CLL/MBL. (**B**) Upset plot of FL samples, FA pathway was more affected in FL. (**C**) Upset plot of CLL/MBL samples, the pathway with most mutations was DDR with 6 cases.

## Data Availability

The sequencing data of follicular lymphoma cases are available from the European Genome-phenome archive (EGA) database under the accession codes EGAD00001001301 and EGAD000010012092. The sequencing data from CLL cases are available at DDBJ under accession number ERP003635. All codes and other data supporting the findings of this study are available at https://github.com/catg-umag/bcell-lymphomas-mutational-signatures.

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
