# Peer review of "Integration of Mutational Signature Analysis with 3D Chromatin Data Unveils Differential AID-Related Mutagenesis in Indolent Lymphomas"

_ijms, 2021, doi:10.3390/ijms222313015_

Round 1

Reviewer 1 Report

The present work describes the distribution of mutations in the main indolent lymphomas of B-cell origin. It unveils also the correlation with the AID motifs that play a role in somatic hypermutation and that are likely to be relevant for these lymphoproliferative malignancies. The specificity is also investigated through the analysis of the TCGA datasets. The data concerning the mutational profiles and their predicted significance are reported.

The work is well presented and the technologies are adequate to describe the chosen topic.

Some concerns are:

  • the limited number of samples.
  • Figure 5: it is rich in information but not too clear. I would put thicker bars in the statistics/n.s. lines in each column. Also, please explain where the differences are statistically significant in the corresponding Results paragraph 3.5(whether is significant in FL or CLL/MBL).
  • Figure 6: in the legend, please specify the meaning of the abbreviations. What is the meaning of MMR pathway?
  • Discussion: it is not clear to me what "de novo decomposition" means (lines 310-316).

Author Response

Point 1: the limited number of samples.
Response 1: We thank the reviewer for raising this point. We have now discussed this limitation in the revised version of the manuscript (lines 312-319): 

On the other hand, the limited number of available samples may restrict a broader extrapolation of our results, however, the sample size remains similar to other studies addressing mutational signatures analysis.

Point 2: Figure 5: it is rich in information but not too clear. I would put thicker bars in the statistics/n.s. lines in each column. Also, please explain where the differences are statistically significant in the corresponding Results paragraph 3.5(whether is significant in FL or CLL/MBL).
Response 2: Figure 5 was edited according to the reviewer suggestion. The bracket lines are thicker now and the statics values (or ns) have a font size increased.

The significance of the statistical information is now added in section 3.5.

Point 3: Figure 6: in the legend, please specify the meaning of the abbreviations. What is the meaning of MMR pathway?
Response 3: We thank the reviewer for noticing this omission. The legend was corrected by adding the description of the acronyms.

Point 4: Discussion: it is not clear to me what "de novo decomposition" means (lines 310-316).
Response 4: We agree with the reviewer, the paragraph in lines 310-316  is not entirely  and was edited as follows (312-319 in the revised manuscript): 

The recently reported signatures SBS84 and SBS85, which are related to canonical AID activity, were difficult to extract from global genomic data. However, these signatures can be readily detected by localized extraction in the IG loci (Supplementary Figure S7A and 5A). This apparent discrepancy indicates that the identification of certain signatures, despite their undisputed presence in the dataset, remains challenging by applying only the de novo signature extraction method and requires the addition of a fitting approach. Of note, the signature SBS9, described as related to non-canonical AID activity, did not emerge within global or localized extraction.

Reviewer 2 Report

This article by  Sepulveda-Yanez et al., is interesting and well written; I have the following minor concerns:

  • The authors need to discuss the limitations of the small sample size used.
  • Section 3.7 should be deleted. The tables and figures should be placed in each section they refer to, in order to make it easier for the reader to follow.

Author Response

Point 1: The authors need to discuss the limitations of the small sample size used.
Response 1: We thank the reviewer for raising this point. We have now discussed this limitation in the revised version of the manuscript (lines 391-393): 

On the other hand, the limited number of available samples may restrict a broader extrapolation of our results, however, the sample size remains similar to other studies addressing mutational signatures analysis.

Point 2: Section 3.7 should be deleted. The tables and figures should be placed in each section they refer to, in order to make it easier for the reader to follow.
Response 2: We thank the reviewer for this comment and followed the suggestion.